# Trends and Pattern of Antibiotic Use in Children in Northern Spain, Interpreting Data about Antibiotic Consumption in Pediatric Outpatients

**DOI:** 10.3390/children9030442

**Published:** 2022-03-21

**Authors:** Laura Calle-Miguel, Carlos Pérez-Méndez, Elisa García-García, Belén Moreno-Pavón, Gonzalo Solís-Sánchez

**Affiliations:** 1Pediatrics Department, Hospital Universitario Central de Asturias, Av. Roma, s/n, 33011 Oviedo, Spain; solisgonzalo@uniovi.es; 2Pediatrics Department, Hospital Universitario de Cabueñes, Camino los Prados, 395, 33394 Gijón, Spain; perezmendez.carlos@gmail.com; 3Pediatric Primary Health Care System, C/ Severo Ochoa s/n, 33208 Gijón, Spain; elisagarciagijon@gmail.com (E.G.-G.); belenmorenop@gmail.com (B.M.-P.)

**Keywords:** pediatrics, antibiotic use, community setting

## Abstract

Monitoring of antibiotic prescription and consumption behavior is crucial. The Access, Watch, and Reserve (AWaRe) classification of antibiotics has been recently introduced in order to measure and improve patterns of antibiotic use. In this study, retrospective data about systemic antibiotic consumption (expressed in defined daily dose per 1000 inhabitants per day (DID)) in pediatric outpatients in a region in northern Spain (around 100,000 children up to 14 years old) from 2005 to 2018 were analyzed and compared with antibiotic consumption in general population in Spain. The pattern of use was analyzed by the percentage of the current AWaRe categories, the Access-to-Watch index, and the amoxicillin index. Data were calculated annually and compared into two periods. Mean antibiotic consumption in pediatric outpatients was 14.0 DID (CI 95% 13.38–14.62). It remained stable throughout the study and was lower than consumption in general population in Spain, particularly from 2016. Changes in the consumption of the main active principles have led to an improvement in the three metrics of the pattern of use. It is important to have a thorough knowledge of the methodology applied in studies about antibiotic consumption. There is a lack of an optimal standardized metric for the pediatric population.

## 1. Introduction

Reducing unnecessary and inappropriate use of antibiotics is a public health priority. Continuous monitoring of antibiotic prescription and consumption behavior is essential to control increasing antibiotic resistance worldwide [1].

The World Health Organization (WHO) launched the Global Antimicrobial Resistance and Use Surveillance System (GLASS) in 2015, which collects data on antibiotic consumption of 65 countries [2]. The European Centre for Disease Prevention and Control (ECDC), through the European Surveillance of Antimicrobial Consumption Network (ESAC-Net), has been monitoring antibiotic consumption in community and hospital sectors in 31 European countries since 1997, describing a high variability between countries. There is a north-to-south and west-to-east increasing gradient in terms of antibiotic consumption and bacterial resistance. Spain is one of the European countries with higher rates of antibiotic consumption and bacterial resistances [3].

The ECDC collects antibiotic consumption information according to the Defined Daily Dose (DDD) methodology developed by the WHO Collaborating Centre for Drug Statistics Methodology. In January 2019, this organism changed the DDDs for several antibiotics [4]. Moreover, the WHO introduced the Access, Watch, and Reserve (AWaRe) classification of antibiotics as part of the updated 2017 Model List of Essential Medicines. Access antibiotics are used as first- or second-line therapies and should be available at an affordable cost (mostly penicillins, sulfonamides, and first-generation cephalosporins), Watch antibiotics are recommended only for specific indications (mostly second and third cephalosporins, fluoroquinolones, and macrolides) and Reserve antibiotics include those of last resort (such as new carbapenems and glycopeptides, fifth-generation cephalosporins, and polymyxins). In 2019, the category of Not Recommended antibiotics was added to the framework, consisting of fixed-dose combinations of multiple broad-spectrum antibiotics that are not evidence based. This classification was developed to measure and drive improvement in antibiotic stewardship efforts on global, regional, and national levels and better understanding of national patterns of antibiotic use. There is a WHO global and national level target for Access antibiotics to account for at least 60% of overall antibiotic consumption by 2023 [5].

Antibiotics are among the most commonly prescribed drugs in pediatrics [6,7]. Nevertheless, units of measure used in most of these monitoring programs are not accurate to control the consumption in this population, as they do not take into account the particularity of children’s dosage regarding the weight [1,8].

This study aims to describe antibiotic consumption and to analyze the pattern of antibiotic use in pediatric outpatients in a region in northern Spain, pointing out difficulties when analyzing data of antibiotic consumption, particularly in children.

## 2. Materials and Methods

### 2.1. Study Setting and Population

Retrospective and descriptive study about consumption of antibacterials for systemic use (J01 group of the Anatomical Therapeutic Chemical Classification, ATC) in pediatric outpatients (up to 14 years old) from 2005 to 2018 in Principado de Asturias, a region in northern Spain, with 101,767 children protected by the National Health System in 2018. Data were collected from the billing data of the containers dispensed in the pharmacy offices run by the National Health System.

### 2.2. Metrics of Antibiotic Consumption and Pattern of Antibiotic Use

Information about antibiotic consumption was expressed as DDD per 1000 inhabitants per day (DID) according to the latest and the previous available ATC/DDD index [9,10]. DIDs were calculated annually and aggregated by J01 group, therapeutic group, and active principle levels. Global data were contrasted to information about antibiotic consumption in Spanish general population by the ECDC [3,9,10].

To analyze the pattern of use, antibiotics were classified into the four current AWaRe categories: Access, Watch, Reserve, and Not Recommended. Antibiotics that could not be assigned to the AWaRe categories were left unclassified categories [5]. Three metrics of pattern of antibiotic use were calculated: first, the percentage of Access, Watch, Reserve, and Not Recommended antibiotic consumption (the DIDs of antibiotics in each group divided by the total DIDs); second, the Access-to-Watch index (the ratio of DIDs of Access antibiotics to Watch antibiotics); and third, the amoxicillin index (the DIDs of amoxicillin divided by the total DIDs) as the recommended first-line therapy for respiratory tract infections, which are the most common indication for antibiotic prescriptions in pediatrics [11,12].

### 2.3. Statistical Analyses

Antibiotic consumption and the pattern of use were calculated annually and compared into two periods (2005–2011 and 2012–2018). From the middle of 2018, an antibiotic stewardship program was implemented in our region, following the guidelines of the National Plan of Antimicrobial Resistance (PRAN) [13]. This program started in the hospital sector, and therefore, our results have not been affected by this strategy. Descriptive analyses were performed using means and confidence intervals of 95% (CI 95%), and variables were compared using the Student’s *t*-test. Statistical analyses were performed using IBM SPSS Statistics software (Statistical Package for the Social Sciences) version 23.0. The statistical significance level was defined as a two-tailed *p*-value < 0.05.

## 3. Results

### 3.1. Antibiotic Consumption

Mean antibiotic consumption in pediatric outpatients in Asturias, a region in northern Spain, between 2005 and 2018 was 14.0 DID (CI 95% 13.38–14.62), showing no statistically significant changes between periods. Figure 1 represents evolution of consumption of antibacterials for systemic use in the pediatric population in the community sector compared to consumption in Spanish general population published by the ECDC according to the previous and the latest ATC/DDD index [9,10].

The most consumed therapeutic groups were J01C (β-lactam penicillins, 10.7 DID; 76.7%), J01F (macrolides, lincosamides, and streptogramins, 1.8 DID; 12.8%), and J01D (other β-lactam antibacterials, referring to cephalosporins, 1.2 DID; 8.9%). The consumption of ten active principles represented 97.6% of the global use, and the consumption of amoxicillin and amoxicillin-clavulanate accounted for 72.8%. Consumption of amoxicillin, phenoxymethylpenicillin, azithromycin, and doxycycline significantly increased, whereas consumption of cefuroxime, cefixime, and clarithromycin decreased throughout the study (Table 1).

### 3.2. Pattern of Antibiotic Use

The consumption of Access antibiotics was greater than 70% every year (minimum of 71.5% in 2005, maximum of 80.4% in 2015; Figure 2), and both Access-to-Watch index and amoxicillin index significantly increased overtime (Table 1).

## 4. Discussion

The present study is the first carried out about antibiotic consumption and pattern of use in the entire pediatric population in our region, placed in northern Spain during a long time period. Furthermore, it shows the differences with antibiotic consumption in the Spanish general population and points out several difficulties when interpreting official data about this topic. The study shows that antibiotic consumption in the pediatric community sector in the study site was lower than consumption in general population in Spain, particularly from 2016, and that the pattern of antibiotic use improved throughout the study.

Readers must be aware of several aspects about interpreting data of antibiotic consumption. First, the ECDC collects antibiotic consumption information according to the DDD methodology developed by the WHO Collaborating Centre for Drug Statistics Methodology [4]. One DDD is the assumed average maintenance dose per day for a drug used in its main indication in a 70 kg adult. DDD is a technical unit of measurement and not a standard for appropriate use, but it makes possible to aggregate information about substances with different pack sizes, strengthening it into units of measurement of active substances. DID represents a standard in performing valid and reliable cross-national or longitudinal studies of antibiotic consumption, and thus, it allows to compare data between regions and countries. DID has been selected as the primary harmonized outcome indicator by the ECDC, the European Food Safety Authority (EFSA), and the European Medicines Agency (EMA) [14,15].

DDD values may be modified over time because of changes in the main indication or amendments for the recommended or prescribed daily dose. In January 2019, the WHO Collaborating Centre for Drug Statistics Methodology changed the DDDs for several antibiotics, including those with the highest consumption in the Spanish community sector: amoxicillin and amoxicillin-clavulanate (both from 1 g to 1.5 g) [4]. Historical ESAC-Net data on antibiotic consumption since 1997 were recalculated accordingly, following the normative to report all data from all years with the latest available ATC/DDD index. Thus, recent data displayed in the interactive database differ from those in historical annual epidemiological reports. Data may appear to be lower at present than previously, with the largest impact for the penicillins group [3,4,9,14]. Readers must be aware of these changes when comparing current data with that published in scientific papers until 2019. We observed a decline around 25% of antibiotic consumption rates in our pediatric population according to the latest ATC/DDD index, similar to the difference observed in the Spanish general population after the ATC/DDD update [9,14,15].

Second, there has been a noticeable increase of antibiotic consumption rates in Spain since 2016. Spain changed the source to report community consumption data from 2016 onwards from reimbursement data to National Health System to sales data, which resulted in a substantial technical increase of antibiotic consumption compared with previous years. The major limitation of reimbursement data is that antibiotics dispensed without a prescription and non-reimbursed prescribed antibiotics are not included. Spanish data are currently more reliable and comparable with those of other countries, as just a quarter of countries reporting information to ESAC-Net still provide reimbursement data [3]. Rates of Spanish antibiotic consumption are worrisome. In 2019, Spain was the fifth European country with the highest antibiotic consumption in the community sector, reaching 23.3 DID. This number is above the European mean (18 DID), while a significant decrease in the mean consumption of antibacterials in Europe was observed over the last ten years [14]. Although prohibited in Spain since 2002, dispensing antibiotics without a medical prescription is still a notorious problem. In simulated scenarios, an antibiotic is sold in 35% to 54% of times when it is requested without prescription [16,17]. Furthermore, there is a high prescription of antibiotics in the private sector in several regions in Spain [13].

Third, prescriptions in children are commonly based on patient weight. Measuring antibiotic consumption by DDD can be found questionable in children, as it quantifies the volume of antibiotic administered but does not take into account body weight. Over 25 different measures can be found to describe antibiotic use in pediatric population, referring to the proportion of exposed patients and quantity and duration of antibiotic therapy [18]. The current recommendation metric in hospitalized patients for pediatric antibiotic stewardship programs is the number of days of therapy (DOT). Antibiotic DDD adjusted by weight has been recently developed for pediatric inpatients; this measure is highly correlated with DOT and names DDD in mg/kg, a standard not defined by the ATC/DDD system [1,8]. These measures have their application in the hospital sector and require patient-level data often inaccessible in studies in the community sector. DDD can still be used to follow antibiotic use density in populations where patients’ average weight is constant, as it could be in an entire pediatric population in a region such as ours [18].

In relation to the pattern of antibiotic use, the majority of non-Access antibiotic consumption in our study was from Watch Antibiotics, with Reserve and Not Recommended antibiotic consumption accounting for less than 0.15%. We observed an improvement in the pattern of antibiotic use in our population in the three mentioned metrics. Azithromycin was the main Watch antibiotic whose consumption increased over time in our study and the active principle with the highest increase in the study region [19]. Recent data on pediatric outpatient antibiotic stewardship suggest focusing on prescriptions of azithromycin in order to decrease antibiotic use [20,21]. Spain achieved the target in the WHO AWaRe framework of at least 60% of Access antibiotics in general population in 2004 and has increased since then up to 68.5% in 2015 [22]. Access antibiotics in our population were above 70% every year. However, our data are still inferior to that presented in the pediatric Spanish population, notifying 90% of Access antibiotics, Access-to-Watch index beyond 10, and amoxicillin index around 0.5 in 2015 [11].

The main current and global problem is the increasing consumption of both Access and Watch antibiotics, with the consumption of Watch antibiotics rising much faster than Access antibiotics, especially in low- and middle-income countries. This challenges the achievement of the AWaRe target for Access antibiotics to account for 60% of the global consumption by 2023 in these countries [11,22,23]. The use of the amoxicillin index can inform the development of initiatives to improve the use of amoxicillin and other Access antibiotics. Countries characterized by high consumption of amoxicillin have national guidelines for the treatment of respiratory tract infections supporting the use of this drug [12]. The AWaRe antibiotic classification provides a useful framework for exploring national antibiotic-consumption patterns [5,11,22,23].

Our study presents several limitations, mostly derived from the analysis and comparison of antibiotic consumption given the already-mentioned particularities of the different populations and measures and changes on the methodology. We did not include data about the hospital sector, below 1.5% of global consumption in our population [19], since the objective was to analyze the consumption and the pattern of antibiotic use in the pediatric community. We did not include data about antibiotic consumption without prescription, estimated at around 8% in the Spanish pediatric population in simulated scenarios [24], or about prescription of antibiotics in the private sector, but we do not expect to have a high rate in our population, as 98% of the pediatric population is covered by the National Health System [19].

## 5. Conclusions

Antibiotic consumption in pediatric outpatients between 2005 and 2018 in the study region, placed in northern Spain, remained stable and was lower than consumption in the general Spanish population. The pattern of antibiotic use improved throughout the study, but some points, such as increasing the amoxicillin index or controlling consumption of some Watch antibiotics, such as azithromycin, need to be improved. There is still a lack of an optimal standardized metric for the pediatric population. The reader must take into account the different methods of measurements and major changes on the data collection methodology used by the main international consumption surveillance systems when interpreting and comparing data about antibiotic consumption.

## Figures and Tables

**Figure 1 children-09-00442-f001:**
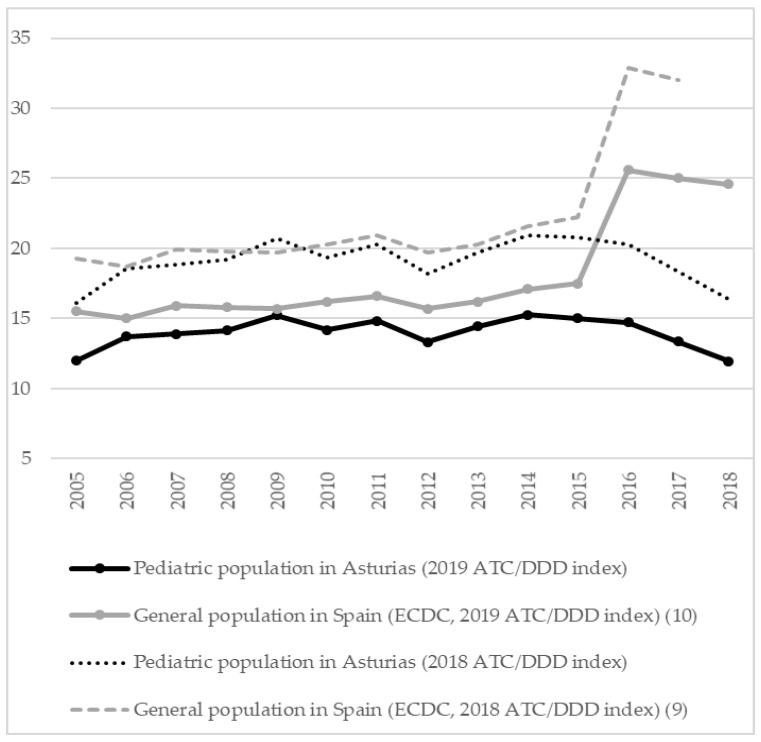
Antibiotic consumption (DID) in the community sector in pediatric outpatients in Asturias and Spanish general population (2005–2018). DID, defined daily dose (DDD) per 1000 inhabitants per day.

**Figure 2 children-09-00442-f002:**
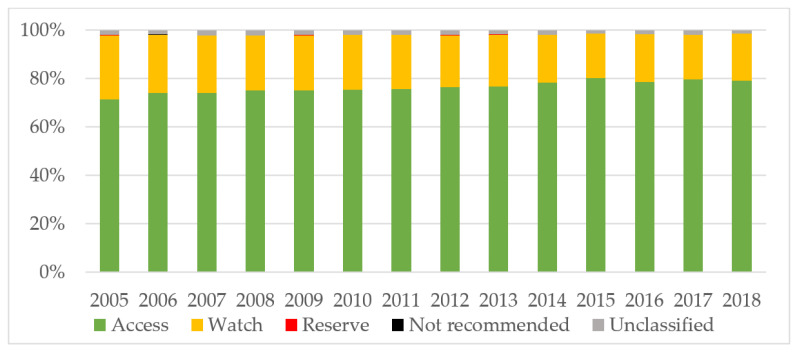
Consumption of antibiotics (%) according to the AWaRe classification in pediatric outpatients in Asturias (2005–2018). AWaRe: Access, Watch, and Reserve antibiotics.

**Table 1 children-09-00442-t001:** Consumption (DID) of total and most commonly used antibiotics and pattern of use in pediatric outpatients in Asturias. Global data and comparison between periods.

Consumption	Code	Global	Period 2005–2011	Period 2012–2018	*p*
Global consumption					
Antibacterials for systemic use	J01	14 (13.38–14.62)	13.99 (13.04–14.95)	14 (12.91–15.1)	0.949
Most commonly consumed antibiotics					
Amoxicillin (Access)	J01CA04	5.16 (4.79–5.52)	4.76 (4.22–5.3)	5.55 (5.18–5.93)	**0.013**
Amoxicillin-clavulanate (Access)	J01CR02	5.03 (4.74–5.32)	5.23 (4.9–5.56)	4.84 (4.30–5.38)	0.180
Azithromycin (Watch)	J01FA10	1.04 (0.86–1.21)	0.79 (0.67–0.91)	1.28 (1.11–1.46)	**0.002**
Cefuroxime (Watch)	J01DC02	0.82 (0.69–0.96)	1.00 (0.93–1.06)	0.65 (0.46–0.84)	**0.004**
Clarithromycin (Watch)	J01FA09	0.61 (0.48–0.73)	0.80 (0.76–0.84)	0.42 (0.30–0.54)	**0.002**
Phenoxymethylpenicillin (Access)	J01CE02	0.33 (0.29–0.36)	0.29 (0.26–0.32)	0.37 (0.32–0.41)	**0.009**
Cefixime (Watch)	J01DD08	0.32 (0.26–0.38)	0.39 (0.35–0.43)	0.25 (0.16–0.34)	**0.009**
Benzathine phenoxymethylpenicillin (Unclassified)	J01CE10	0.21 (0.19–0.24)	0.2 (0.16–0.24)	0.23 (0.19–0.26)	0.277
Doxycycline (Access)	J01AA02	0.09 (0.06–0.11)	0.05 (0.04–0.06)	0.12 (0.1–0.15)	**0.002**
Trimethoprim/Sulfamethoxazole (Access)	J01EE01	0.06 (0.05–0.06)	0.06 (0.05–0.07)	0.06 (0.05–0.06)	0.949
Pattern of antibiotic use					
Access antibiotics (%)		76.47% (75.02–77.92)	74.46 (73.11–75.81)	78.48 (77.12–79.84)	**0.002**
Access-to-Watch index		3.58 (3.29–3.88)	3.18 (2.95–3.4)	3.99 (3.67–4.30)	**0.002**
Amoxicillin index		0.37 (0.35–0.39)	0.34 (0.32–0.36)	0.4 (0.38–0.41)	**0.002**

Data presented correspond to means and confidence interval of 95%. Numbers in bold font: statistically significant differences (*p* < 0.05). DID, defined daily dose (DDD) per 1000 inhabitants per day.

## Data Availability

Restrictions apply to the availability of these data. Data were obtained by a permission from the Sub-Direction of the Organization of Health System of Principado de Asturias. All the authors have full access to the data, and L.C.-M. is the guarantor for the data.

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
