# Peer review of "Trends and Pattern of Antibiotic Use in Children in Northern Spain, Interpreting Data about Antibiotic Consumption in Pediatric Outpatients"

_children, 2022, doi:10.3390/children9030442_

Round 1

Reviewer 1 Report

The article is well conceived and provides useful information on antibiotic use in Spanish children. Monitoring of antibiotic use over the years is crucial due to the rise in antibiotic resistance in children all over the world.

Minor English grammar checks are needed - e.g. line 165, "with that" must be corrected into "with those" being "data" a plural noun.

Other than that, I suggest publication with minor editing needed.

Author Response

Dear reviewer of Children,

In response to the revision of the manuscript entitled “Trends and pattern of antibiotic use in children in Northern Spain. Interpreting data about antimicrobial consumption in outpatients”, we would like to thank for your valuable suggestions that will certainly help to improve the quality and understanding of the manuscript. We checked thoroughly the text and English grammar and other editing changes have been done. Moreover, we have corrected some punctuation marks and several mistakes in the order of the bibliographical references. The manuscript has been checked and approved by all authors, who accept full responsibility for the content. We hope that the editorial board and the reviewers will agree on the performed changes of the manuscript.

Reviewer 2 Report

Te authors describe antibiotic consumption in children in the community in Nortern Spain from 2005-2018.The consumption is expressed in DIDs using ATC/DDD, version 2019.

My comments:

  • Not recommended antibiotics should be mentioned in short in the introduction or later among results
  • -Many countries publish national consumption data in children in number of presctriptions per 1000 children/year. Could the authors present additionally the total consumption in this metric? (for example Cizman M.et al.J Pediatr Inf Dis 2014;9:1-7)
  • Could the authors assess the OTC antibiotic consumption in their region more exactly?
  • Were in the region any stewardship actions implemented during whole period?
  • .In the discussion comparison with other countries should be described 

.

Author Response

Dear reviewer of Children,

In response to the revision of the manuscript entitled “Trends and pattern of antibiotic use in children in Northern Spain. Interpreting data about antimicrobial consumption in outpatients”, we would like to thank for your valuable suggestions that will certainly help to improve the quality and understanding of the manuscript. Please find below your comments followed by our responses.

  • Not recommended antibiotics should be mentioned in short in the introduction or later among results

Following this recommendation, definitions of Not Recommended antibiotics have been included in short in the introduction, as well as main antibiotics from other categories for a better understanding of the AWaRe classification.

  • Many countries publish national consumption data in children in number of prescriptions per 1000 children/year. Could the authors present additionally the total consumption in this metric? (for example Cizman M.et al.J Pediatr Inf Dis 2014;9:1-7)

The authors agree with the reviewer in the interest of knowing the antibiotic consumption in different measures. However, that there are more than 25 different measures to describe antibiotic use, each one with its advantages and disadvantages, specifically in the pediatric population. The number of prescriptions per 1000 children/year have the disadvantage of referring to substances with different DDDs, different pack sizes and formulations, for a variable length of treatment and can be affected by changes in the commercial presentations overtime. Being the purpose of the study to describe the antibiotic consumption and the pattern of use in a population over a long period of time, comparing information with national recent data and pointing out the difficulties of measurement in the pediatric population, we decided to use the official, standard and recent methods, such as DID and AWaRe system. Nevertheless, our group is working in comparing the antibiotic consumption in our pediatric population in recent years by different measures to know the implications of different methods and we hope to publish our data soon.

  • Could the authors assess the OTC antibiotic consumption in their region more exactly?

We have added information about Over-The-Counter prescription of antibiotics in general and pediatric population in Spain, a still notorious problem in our country that deserves to be mentioned.

  • Were in the region any stewardship actions implemented during whole period?

We have included information about an antibiotic stewardship program implemented in our region in 2018 that did not affect our results.

  • In the discussion, comparison with other countries should be described.

 We really appreciate the reviewer’s suggestion of comparing our data with those from other countries. Nevertheless, measures of antibiotic consumption vary widely in the literature and there are many factors that determine such a variable consumption among areas, even in the same country.  Thus, we have preferred to compare only with national data and focus in explaining the problems when interpreting data about antibiotic consumption, especially in the pediatric population, and to allow the reader to have information about comparable and official data in the European database, already mentioned in the bibliography.

The manuscript has been checked and approved by all authors, who accept full responsibility for the content. We hope that the editorial board and the reviewers will agree on the performed changes of the manuscript.

Round 2

Reviewer 2 Report

Thank you for your reply. The changes iimproved the quality of the paper.

Author Response

Thank for your reply. 

We hope that the editorial board will agree on the performed changes of the manuscript.